# Scrub and murine typhus seroprevalence among blood donors in Laos

Weerawat Phuklia[1*], Jantana Wongsantichon[2,3], Chantala Souksakhone[4], Ampai Tanganuchitcharnchai[3], Mavuto Mukaka[2,3], Kaisone Padith[1], Koukeo Phommasone[1], Mayfong Mayxay[1,2,5], Stuart D. Blacksell[2,3], Audrey Dubot-Pérès[1,2,6], Matthew T. Robinson[1,2], Elizabeth A. Ashley[1,2]

1 Lao-Oxford-Mahosot Hospital-Wellcome Trust Research Unit (LOMWRU), Microbiology Laboratory, Mahosot Hospital, Vientiane, Lao People's Democratic Republic, 2 Centre for Tropical Medicine & Global Health, Nuffield Department of Medicine, University of Oxford, Oxford, United Kingdom, 3 Mahidol-Oxford Tropical Medicine Research Unit, Faculty of Tropical Medicine, Mahidol University, Bangkok, Thailand, 4 Blood Donation Centre, Laos Red Cross, Vientiane, Lao People's Democratic Republic, 5 Institute of Research and Education Development (IRED), University of Health Sciences, Ministry of Health, Vientiane, Lao People's Democratic Republic, 6 Unité des Virus Émergents (UVE: Aix-Marseille Univ, Università di Corsica, Marseille, France

* weerawat@tropmedres.ac

## Abstract

### Background

Scrub typhus and murine typhus, caused by *Orientia tsutsugamushi* and *Rickettsia typhi,* respectively, are important causes of febrile illness in Laos. Although several studies have assessed rickettsial infection in selected provinces, the nationwide distribution remains unclear. This study aimed to estimate exposure to scrub typhus group (STG) and typhus group (TG) across twelve provinces of Laos and identify potential hotspots.

### Methodology

We screened 1,200 serum samples from blood donors (100 per province) for STG and TG IgG antibodies using enzyme-linked immunosorbent assays (ELISA). Samples with optical density ≥ 0.5 were confirmed by immunofluorescence assays (IFA).

### Results

Overall seroprevalence was 7.26% (95%CI:5.93-8.87) for STG and 4.09% (95% CI:3.11-5.37) for TG. STG seroprevalence was highest in Huaphan (27%), Oudomxay (19%) and Xiangkhuang (17%), all in northern Laos. TG seroprevalence was 10% in both Oudomxay (north) and Attapue (south). Compared with Vientiane Capital, these provinces had significantly higher seropositivity. No significant association was observed with age group. STG seropositivity was higher in males, whereas TG seropositivity was higher in females.

**Data availability statement:** All data underlying the findings of this study are fully available without restriction. The raw dataset containing patient ID codes, demographic information, ELISA OD values, IFA titres, interpreted serology results, and the associated data dictionary is included as Supporting Information (S1 Dataset). All additional summary tables and analyses are available within the manuscript and Supporting Information files.

**Funding:** This research was funded in whole, by the Wellcome Trust [220211/Z/20/Z] to EAA. WP and KaP were supported by the Wellcome Trust through an International Training Fellowship (220690/Z/20/Z to WP). For the purpose of Open Access, the author has applied a CC BY public copyright license to any Author Accepted Manuscript version arising from this submission. The funders had no role in study design, data collection and analysis, decision to publish, or preparation of the manuscript.

**Competing interests:** I have read the journal's policy and the authors of this manuscript have the following competing interests: SDB is a Section Editor for PLOS Neglected Tropical Diseases. This does not alter our adherence to PLOS policies on sharing data and materials. The authors declare that no other competing interests exist.

## Conclusion

Rickettsial infections are widespread in Laos, with STG seroprevalence concentrated in the north and TG seroprevalence present in both the north and south, guiding future research priorities and informing targeted public health interventions.

## Author summary

Scrub typhus and murine typhus are caused by bacteria that can only survive inside cells, *Orientia tsutsugamushi* and *Rickettsia typhi*, respectively. These bacteria are transmitted to humans by arthropods: chigger mites for scrub typhus and fleas for murine typhus. Both diseases cause treatable fever, with symptoms ranging from mild to severe. Although exposure to these infections has been reported in Laos, most studies have been limited to central areas and a few provinces, which may not represent the entire country. This study investigated whether healthy blood donors across Laos had prior exposure to scrub typhus and murine typhus. We screened antibodies against both pathogens from 100 samples per province, collected from 12 provinces in Laos. We found that individuals in northern Laos, particularly in Huaphan, Oudomxay, and Xingkhuang, were more frequently exposed to *O. tsutsugamushi* compared to individuals in other provinces. In contrast, exposure to *R. typhi* was observed in both northern and southern Laos. However, no antibodies against scrub typhus were detected in our selected population in Salavan, despite the fact that previous studies of febrile patients have reported laboratory-confirmed infections, and no antibodies against murine typhus were found in Huaphan. This study suggests that scrub typhus and murine typhus are present throughout Laos, with varying patterns of exposure across different regions, although blood donors may be a low-risk population in terms of exposure.

## Introduction

*Rickettsia* bacteria are obligate intracellular, Gram-negative coccobacilli transmitted by arthropod vectors. The rickettsioses are divided into three major antigenic groups: the typhus group (TG), the spotted fever group (SFG), and the scrub typhus group (STG). In Laos, the main rickettsial pathogens causing human disease are *Orientia tsutsugamushi* and *Rickettsia typhi*, which are transmitted to humans by infected chigger mites and fleas, respectively. *O. tsutsugamushi* is the causative agent of scrub typhus, while *R. typhi* causes murine (endemic) typhus. Both diseases are important causes of acute febrile illness in Laos [1].

Rickettsial infections typically present with acute fever, often accompanied by headache, myalgia, nausea, vomiting, and rash. The overall mortality rate from rickettsial infections varies by species, ranging from as low as 0.4% for murine typhus [2] to approximately 6% for scrub typhus in untreated patients [3]. Mortality rates are

higher when patients develop complications such as meningitis or meningoencephalitis. Both scrub typhus and murine typhus have been identified as causes of central nervous system infection in Laos [4]. The presence of an eschar is a useful clinical sign for distinguishing scrub typhus from murine typhus and from other febrile illnesses, such as dengue fever, which often present with similar symptoms [5]. Moreover, bacterial loads in scrub and murine typhus patient blood are typically low, limiting the effectiveness of antigen-based diagnostic methods, including qPCR, which often lack sensitivity for detecting infections in patients with mild disease.

Although the immunofluorescence assay (IFA) using paired sera remains the diagnostic gold standard for comparing IgM and IgG titers, the requirement for convalescent serum delays confirmation. Previous studies reported overall seroprevalence rates of 20.3% for STG IgG and 20.6% for TG IgG in areas around Vientiane Capital, although positivity varied greatly across the survey area. STG positivity was higher in rural outskirts of the city, whereas TG positivity was higher in the city center. Although both TG and STG have been reported throughout Laos, little is known about their true distribution, which likely varies considerably across towns and provinces. Screening blood donors can provide a clearer picture of rickettsial distribution in Laos and the potential risks of *Rickettsia* infection in the context of blood transfusion safety, acknowledging that blood donors may not have the highest exposure risk, particularly to *O. tsutsugamushi*.

The aim of this study was to estimate the burden of rickettsioses, focusing on scrub typhus and murine typhus, across different provinces in Laos. This seroprevalence study tested IgG against STG and TG in stored serum from blood donors in 12 provinces using ELISA and IFA. In addition, Geographic Information System (GIS) mapping was used to illustrate the spatial distribution of cases.

## Methods

### Ethics statement

The samples in this study were stored serum samples from blood donors in Laos collected by the Lao National Blood Centre from 12 out of the 17 provinces in Laos. Dates of donation were between January 2021 and August 2021. Blood donors gave written informed consent for leftover samples to be used for future research at the time of blood donation. Ethical approval for the study was granted by the University of Health Sciences Ethics Committee, Vientiane, Laos and the Oxford Tropical Research Ethics Committee (OxTREC 52–20).

### Blood donors

The study population was stratified into five age groups (17–24, 25–30, 31–36, 37–46, and ≥47 years). These categories were selected to capture distinct life stages, ranging from late adolescence and young adulthood to mid-life and older adulthood. The chosen ranges also ensured adequate sample sizes in each group, enabling robust statistical analysis while maintaining relevance to sociocultural and occupational transitions that may influence exposure risk. Serum samples were collected at the National Blood Centre and stored at -30°C before being transferred to the Lao-Oxford-Mahosot Hospital-Wellcome Trust Research Unit (LOMWRU), where they were stored at -80°C. For the detection of IgG antibodies against the scrub typhus group (STG) and typhus group (TG), we selected 100 serum samples from each of twelve provinces in Laos (total = 1200).

Most blood donors are younger adults and random sampling was not possible. Selection of samples for this study was based on donor age and sample availability, to be representative of different age groups. Each sample was thawed once to create aliquots and then refrozen at -80°C. Once all 1,200 samples were selected and prepared, they were transported on dry ice from Vientiane to Mahidol-Oxford Tropical Medicine Research Unit (MORU) in Bangkok, Thailand for laboratory analysis.

### Laboratory analysis

IgG antibody testing for rickettsial infections was conducted to estimate the prevalence of prior infections in the population. All samples were screened for IgG antibodies against *O. tsutsugamushi* strains (Karp, Kato, Gilliam and TA716) and

*R. typhi* strain Wilmington using antigens provided by the US Naval Medical Research Center (NMRC), with in-house ELISAs developed at MORU. These ELISAs for *O. tsutsugamushi* (STG) and *R. typhi* (TG) were based on the NMRC ELISA protocol for rickettsial infection [6,7].

Ninety-six-well microtiter ELISA plates were prepared by coating half of each plate with antigens for either STG and TG, diluted in phosphate-buffered saline (PBS) at concentrations of 1:8000, 1:4000 and 1:2000 dilutions for each rickettsial group, respectively, at 100 µL/well. Intracellular bacteria were isolated from infected cell cultures and subjected to sonication to generate whole-cell lysate antigens. Because direct quantification of bacteria protein concentration was not feasible, antigen coating concentrations were determined the optimal working dilution. The other half of the plate was coated with PBS alone as a background control. Coated plates were covered and incubated in a humidified chamber at 4 °C for 36–48 hours.

After incubation, plates were blocked with 200µL/well of blocking buffer (5% skimmed milk in wash buffer: 0.1% Tween 20 in PBS). Serum samples were diluted 1:100 in blocking buffer, added at 100 µL/well, and incubated for 1 hour in a moist chamber at room temperature. Plates were then washed four times with wash buffer. Subsequently, horseradish-peroxidase-conjugated goat anti-human IgG (Invitrogen Corp., Carlsbad, CA, USA) was added at a 1:1000 dilution in blocking buffer (100 µL/well) and incubated for 1 hour at room temperature in a moist chamber.

Following incubation with the secondary antibody, the plates were washed four times with wash buffer. Next, tetramethylbenzidine substrate (KPL Inc., Gaithersburg, MD, USA; 100 µl/well) was added and incubated for 15 min at room temperature in the dark. Finally, 100 µL/well of 1 M hydrochloric acid was added to each well to stop the reaction. Plates were read at 450 nm using a microplate reader (Multiskan FC; ThermoFisher Scientific, Waltham, MA, USA) with the optical densities (ODs) from wells without antigen subtracted as background absorbance to generate a final net OD. Negative and positive control samples were used to control assay performance and were included in four wells each on each plate. If a sample had an ELISA IgG OD ≥ 0.5 [8], subsequent IgG IFA was performed for the same rickettsia group, using the same antigens. Bound antibodies were detected using goat anti-human IgG (Fc)-specific fluorescein isothiocyanate (FITC) conjugate (SeraCare USA; KPL 02-10-20; catalog no. 5230–0291), diluted 1:100 in PBS containing skim milk and 0.05% Evans blue counterstain. The IFA determined the IgG antibody titers by serially diluting samples two-fold from 1:100–1:25600. The endpoint titer was defined as the highest dilution that showed specific fluorescence, as described previously [9,10]

### Definition of seropositivity

In this study, seropositivity was defined as IFA IgG titre ≥ 1:100 following initial screening by ELISA with an optical density (OD) ≥ 0.5, indicating previous exposure to a rickettsial infection [9,11].

### Data analysis

Blood donor characteristics were described using numbers and percentages for gender and age groups, and median with interquartile range (IQR) for age. Fitted linear regression lines were plotted for age (in years) against ELISA IgG OD. Prevalence and 95% confidence intervals (CI) for STG and TG IgG seropositivity, based on ELISA positivity and subsequent IFA confirmation, were calculated using Wilson score intervals. Logistic regression (both unadjusted and adjusted) was used to compute odds ratios (ORs) and 95% CIs to assess the association between participant characteristics (gender, age per 10 years, and region) and the odds of STG and TG IgG seropositivity by IFA. All participants with ELISA OD < 0.5 were not tested by IFA and were considered to be seronegative. The region with the highest number of participants with an IFA titre ≥ 1:100 for both the STG and TG groups was used as the reference region. Data management and analysis were performed using RStudio, (version 2024.12.1 + 563) and GraphPad Prism 10.4.1. Maps were generated in R using an administrative boundary shapefile downloaded from the geoBoundaries Global Administrative Database (Lao PDR country file; https://www.geoboundaries.org/countryDownloads.html), licensed under CC BY 4.0.

## Results

### Demographic characteristics of blood donors from 12 provinces in Laos

A total of 1,200 donors were selected as described above from retrospective blood donor samples collected across twelve out of seventeen provinces, as shown in the map (Fig 1). Two donors from Salavan Province were excluded due to missing age and gender information. Therefore, demographic data were available for 1,198 donors. Their characteristics by region are presented in Table 1. The median age was 31 years (IQR 25–37 years; range 17–69 years) and the majority of blood donors were male (68.70%).

### ELISA screening

All donor samples (n = 1198) were screened using ELISA for STG and TG IgG antibodies. The overall median OD percentage for STG was 0.032 (0.015, 0.131), and for TG it was 0.046 (0.020, 0.162) (S1 Table). Donors with an ELISA OD ≥ 0.5 for STG accounted for 118 participants (10.20%, 95% CI: 8.56–12.10), while 139 participants (12.10%, 95% CI: 10.30-14.10) had ELISA OD ≥ 0.5 for TG (S2 Table). We identified seven participants (0.58%), each of whom had an ELISA OD ≥ 0.5 for both STG and TG.

Among participants with an ELISA OD ≥ 0.5 for STG, when categorized by gender, we observed ELISA positivity in 11.91% of males (95% CI: 9.87–14.3) and 5.33% of females (95% CI: 4.48–8.09) (S2 Table and Fig 2A). The highest

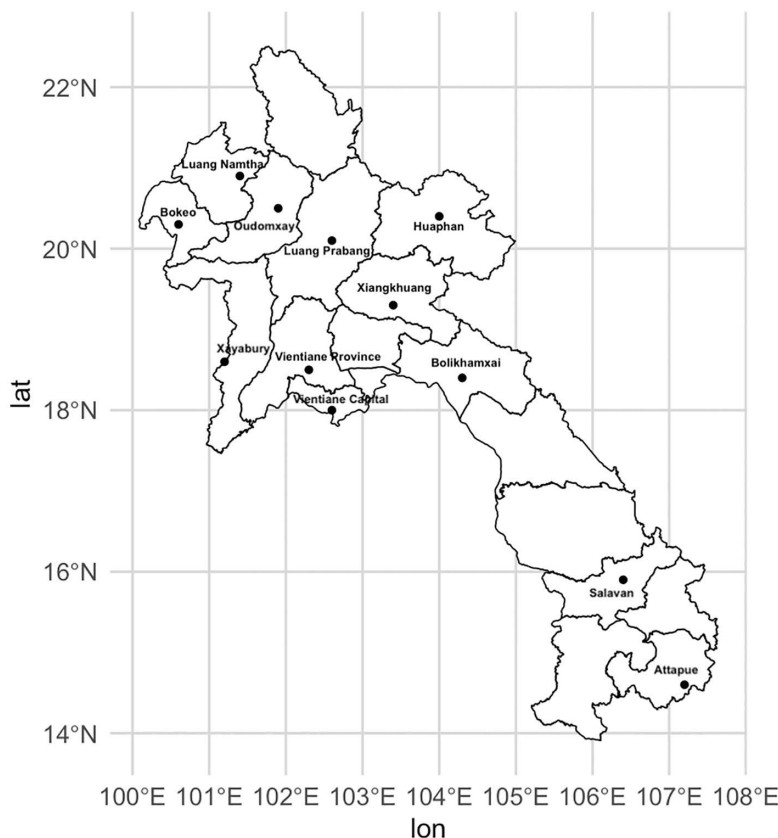

**Fig 1. Twelve provinces (black dot) in Laos where serum samples from blood donors were screened for IgG antibodies against STG or TG.** Maps were generated in R using an administrative boundary shapefile downloaded from the geoBoundaries Global Administrative Database (Lao PDR country file; https://www.geoboundaries.org/countryDownloads.html), licensed under CC BY 4.0.

**Table 1. Demographic characteristics of blood donors in Laos.**

| Provinces | n | Male gender, % | Median age (IQR), years | % of participants by age group, years | | | | |
|---|---|---|---|---|---|---|---|---|
| | | | | 17-24 | 25-30 | 31-36 | 37-46 | ≥47 |
| Attapue | 100 | 66 | 32 (28-38) | 15 | 26 | 26 | 27 | 6 |
| Bokeo | 100 | 77 | 30 (25.8-33) | 22 | 36 | 28 | 13 | 1 |
| Bolikhamxai | 100 | 71 | 22.5 (18-32) | 52 | 18 | 14 | 14 | 2 |
| Huaphan | 100 | 70 | 32 (28-37.2) | 10 | 29 | 33 | 23 | 5 |
| Luang Namtha | 100 | 67 | 31 (27-36.2) | 8 | 41 | 26 | 22 | 3 |
| Luang Prabang | 100 | 62 | 25 (19-35) | 49 | 16 | 13 | 13 | 9 |
| Oudomxay | 100 | 70 | 36 (34-38.2) | 0 | 1 | 50 | 46 | 3 |
| Vientiane Capital | 100 | 60 | 30 (24-38.2) | 26 | 29 | 18 | 19 | 8 |
| Vientiane Province | 100 | 69 | 33 (29-38) | 3 | 32 | 33 | 26 | 6 |
| Salavan | 98 | 68.37 | 28.5 (20-37.8) | 36.73 | 17.35 | 16.32 | 23.46 | 6.12 |
| Xayabury | 100 | 71 | 34 (31-38) | 3 | 12 | 51 | 31 | 3 |
| Xiangkhuang | 100 | 73 | 24 (20.8-32) | 55 | 15 | 15 | 13 | 2 |
| Total | 1198 | 68.70 | 31 (26-37) | 23.29 | 22.70 | 26.96 | 22.54 | 4.51 |

ELISA positivity for STG was found in participants aged 47 years or older (13%, 95% CI: 6.42-24.4), while the lowest was observed in those aged 25–30 years (8.46%, 95% CI: 5.7-12.40) (S2 Table and Fig 2C). When categorized by province, the highest ELISA positivity for STG was observed in Xiangkhuang (31%, 95% CI: 22.8–40.6), followed by Huaphan (30%, 95% CI: 21.90–39.60) and Oudomxay (22%, 95% CI: 15–31.10) (S2 Table and Fig 2E).

Similarly, for TG, ELISA positivity was observed in 11.66% of males (95% CI: 9.65–14.0) and 11.47% of females (95% CI: 8.63–15.10) (S2 Table and Fig 2B). The same age-related pattern was observed, with the highest positivity in participants aged 37–46 years (17%, 95% CI: 13–22) and the lowest in those aged was aged equal or more than 17–24 years (4.66%, 95% CI: 2.74–7.81) (S2 Table and Fig 2D). The top three provinces with the highest ELISA positivity for TG were Oudomxay (27%, 95% CI: 19.30–36.40), Attapue (20%, 95% CI: 13.30–28.90), and Vientiane Province (18%, 95% CI: 11.70-26.70) (S2 Table and Fig 2F).

## Seropositivity estimated by IFA

Among the 1,198 samples screened, 7.26% (95% CI: 5.93-8.87) were seropositive for STG (IFA titer ≥ 1:100) and 4.09% (95% CI: 3.11-5.37%) were seropositive for TG (IFA titer ≥ 1:100) (S3 Table). Among 85 donors seropositive for STG, high antibody titres (≥ 1:400) were detected in 48 donors (56.47%). In contrast, high antibody titres for TG (≥ 1:400) were found in 7 donors (14.58%). The highest titre for STG group was 1:3200, whereas the highest titre for TG was 1:800 (Fig 3).

The percentages of seropositive samples detected by IFA for scrub typhus group (STG) and typhus group (TG), along with 95% CI stratified by gender, age group and province are presented in S3 Table and Fig 4. For STG IgG seropositivity, the positivity rate was 8.87% in males and 3.73% in females (Fig 4A). STG IgG seropositivity by age group was 5.38%, 5.88%, 8.98%, 8.52%, and 7.41% for ages 17–24, 25–30, 31–36, 37–46, and ≥47 years, respectively (Fig 4B). By province, STG IgG seropositivity was 27% in Huaphan, 19% in Oudomxay, and 16% in Xiangkhuang. In contrast, no STG IgG antibodies were detected among participants from Salavan (0%) (Fig 4C and 4D).

For TG IgG seropositivity, the positivity rate was 5.87% in females and 3.28% in males (Fig 4A). TG IgG seropositivity by age group was 1.79%, 4.41%, 4.33%, 5.93%, and 3.70% for ages 17–24, 25–30, 31–36, 37–46, and ≥47 years, respectively (Fig 4B By province, TG IgG seropositivity was 10% in Attapue and Oudomxay, 8% in Bokeo, and 4% in Luang Namtha, Luang Prabang, and Vientiane Province, with other provinces showing lower or no seropositivity. No TG IgG seropositivity was detected in participants from Huaphan (Fig 4C and 4E).

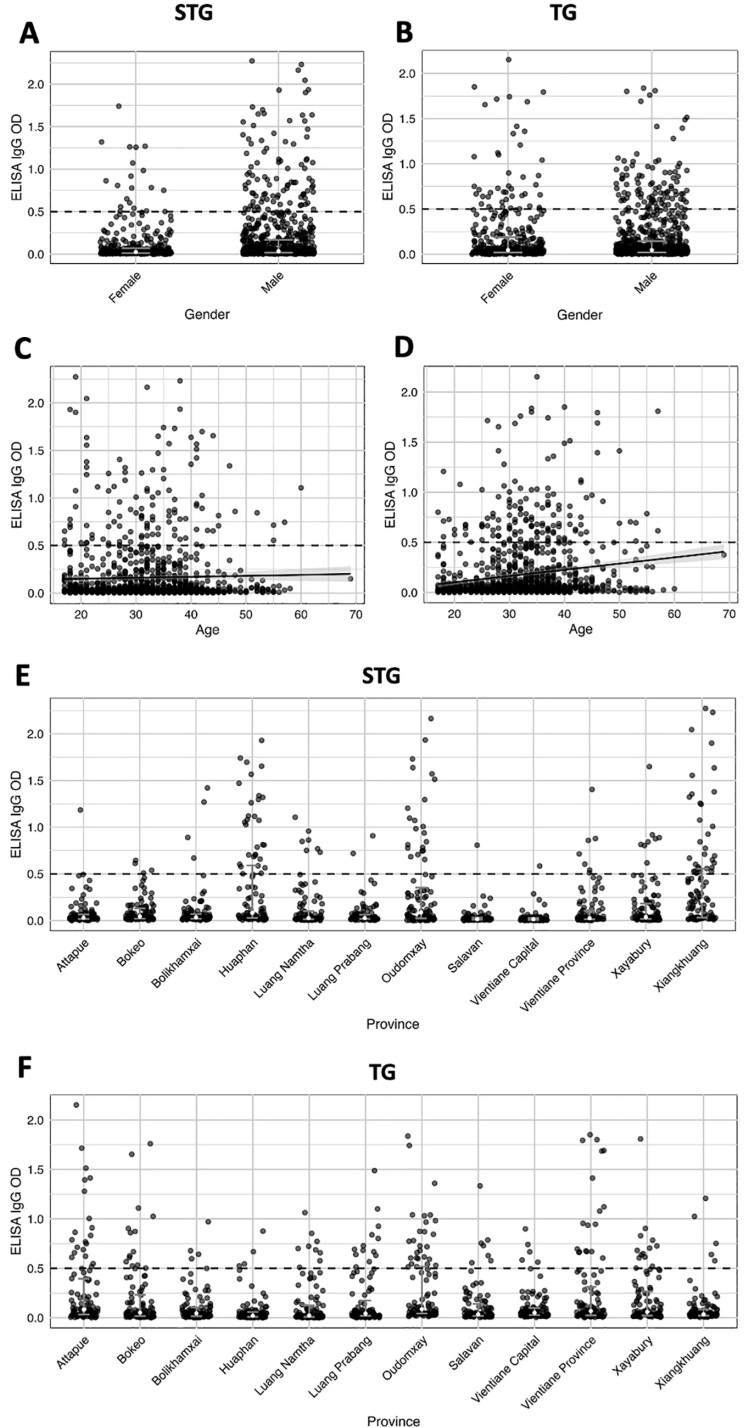

**Fig 2. Optical densities (ODs) of enzyme-linked immunosorbent assay (ELISA) for gender, age in years, and province for scrub typhus group (A, C and E) and typhus group (B, D, and F).** The white dots represent median of OD with interquartile range. The dashed line for gender (A and B), for age (C and D), and for province **(E and F)** indicates ELISA OD ≥ 0.5, which was used as the cut-off point for further testing with immunofluorescence assay (IFA). The line of OD scatter plot by age (C and D) is a fitted linear regression line between age and ELISA IgG OD.

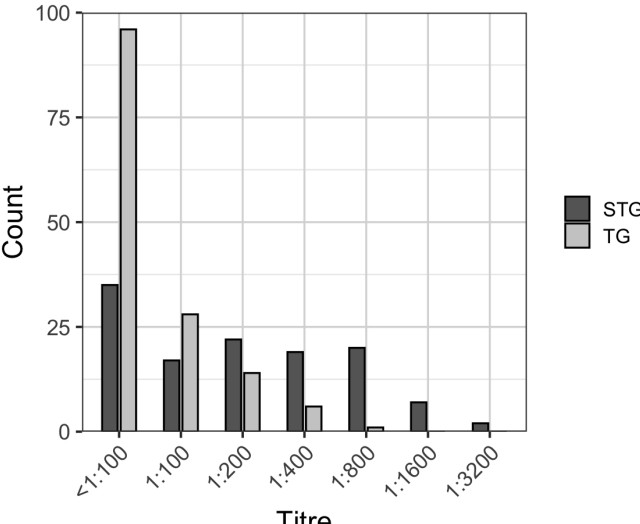

**Fig 3. Immunofluorescence assay (IFA) titres of all samples with enzyme-linked immunosorbent assay optical density ≥ 0.5 for scrub typhus group (STG) represented as the dark grey bar and typhus group (TG) represent as the light grey bar.**

Although seven samples were ELISA-positive for both TG and STG, confirmatory testing by IFA demonstrated that none were positive for both infections.

## Logistic regression and odds of seropositivity

Logistic regression was performed to estimate the association between participant characteristics (gender, age group, and province) and the odds of seropositivity.

For the STG group (Fig 5 and S4 Table), males had higher odds of seropositivity than females, both unadjusted (OR 2.51, 95% CI: 1.44–4.69) and adjusted (aOR 2.44, 95% CI: 1.36-4.68). Age group was not significantly associated with STG seropositivity in either unadjusted or adjusted analyses. By province (reference: Vientiane Capital), Huaphan (OR: 36.62, 95% CI: 7.53–660.62; aOR: 35.96, 95% CI: 7.31–650.99), Oudomxay (OR: 23.22, 95% CI: 4.66–421.71; aOR: 21.87, 95% CI: 4.22–402.45), and Xiangkhuang (OR: 18.86, 95% CI: 3.73–343.90; aOR: 18.17, 95% CI: 3.54-333.04) showed the highest odds ratios. Odds ratios could not be calculated for Salavan due to the absence of positive cases.

For the TG group (Fig 5 and S5 Table), males had lower odds of seropositivity than females, both unadjusted (OR: 0.54, 95% CI: 0.31–0.98) and adjusted (aOR: 0.53, 95% CI: 0.29–0.96). Age group was not significantly associated with TG seropositivity. By province (compared with Vientiane Capital), Attapue (OR: 5.44, 95% CI: 1.39–36.04; aOR: 5.48, 95% CI: 1.38–36.54), Oudomxay (OR: 5.44, 95% CI: 1.39–36.04; aOR: 5.13, 95% CI: 1.23–35.31), and Bokeo (OR: 4.26, 95% CI: 1.03–28.72; aOR: 4.76, 95% CI: 1.13–32.43) had the highest odds. Odds ratios could not be calculated for Huaphan due to the absence of positive cases. Several odds ratios for both STG and TG had wide confidence intervals, indicating substantial uncertainty around these estimates.

## Discussion

This study aimed to investigate the seroprevalence of prior exposure to rickettsial infections, particularly *O. tsutsugamushi* (STG) and *R. typhi* (TG) groups, using retrospective serum samples collected from blood donors across twelve provinces in Laos. IFA results showed the presence of STG IgG in approximately 7% of participants, and TG-specific IgG antibodies in around 4%. Males were more likely to have been exposed to STG than females, while females showed a

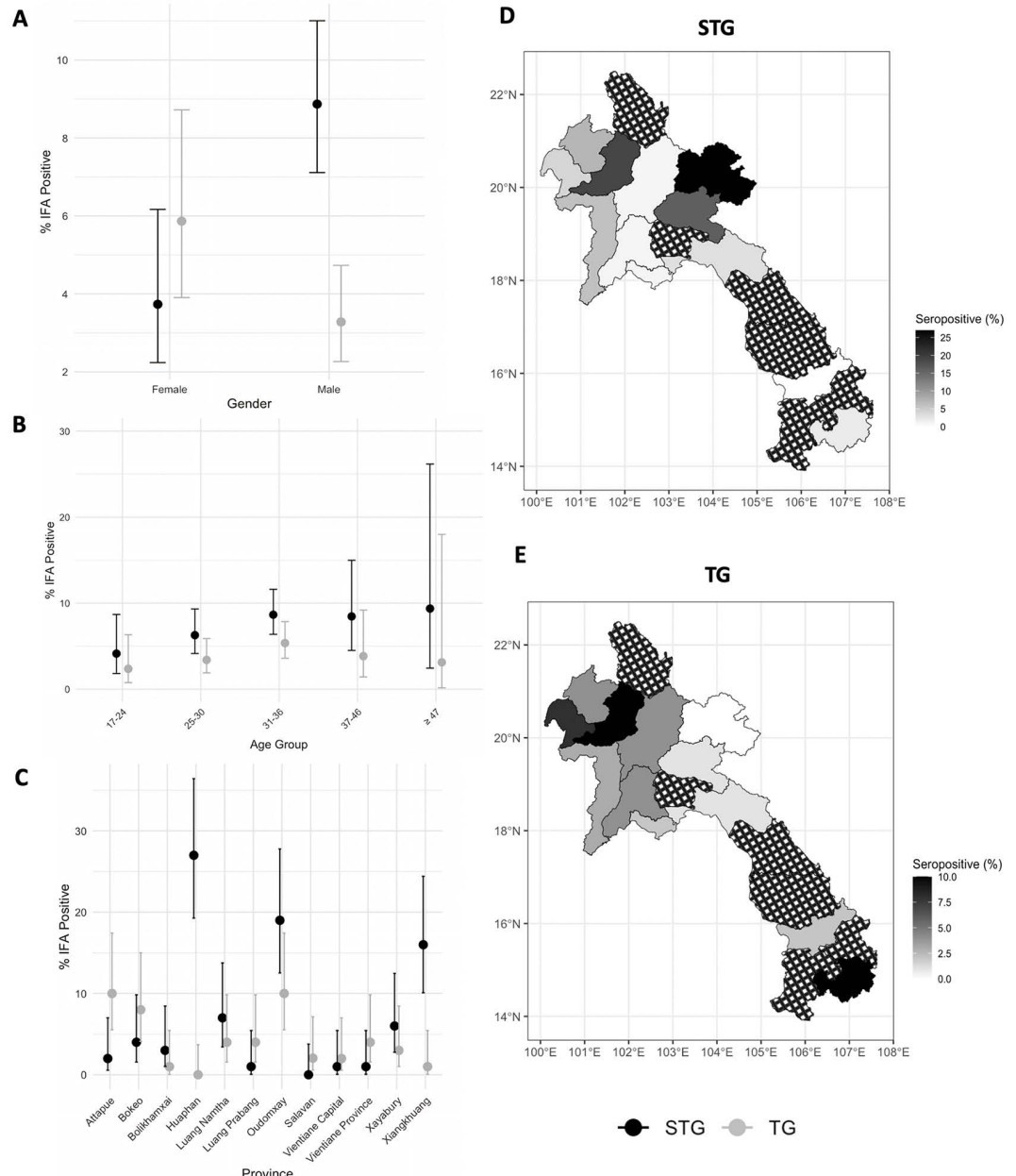

**Fig 4. Percentage and 95% confidence intervals of seropositive participants out of all participants categorized by gender (A), age group (B) and provinces (C) with map (D, E) for scrub typhus group (STG) and typhus group (TG), respectively.** Maps were generated in R using an administrative boundary shapefile downloaded from the geoBoundaries Global Administrative Database (Lao PDR country file; https://www.geoboundaries.org/countryDownloads.html), licensed under CC BY 4.0. Seropositivity was defined as immunofluorescence assay (IFA) titre ≥ 1:100 following enzyme-linked immunosorbent assay (ELISA) optical density (OD) ≥ 0.5. All participants with ELISA OD < 0.5 were not tested with IFA and were considered to be seronegative. Areas with crosshatched shading represent non-investigated regions.

higher prevalence of TG exposure. An explanation for why murine typhus seropositivity was higher in females is that it is typically associated with peridomestic rodent-fleas transmission in urban and peri-urban environments. This finding may reflect differences in patterns of household or peridomestic exposure; however, we did not collect individual-level data

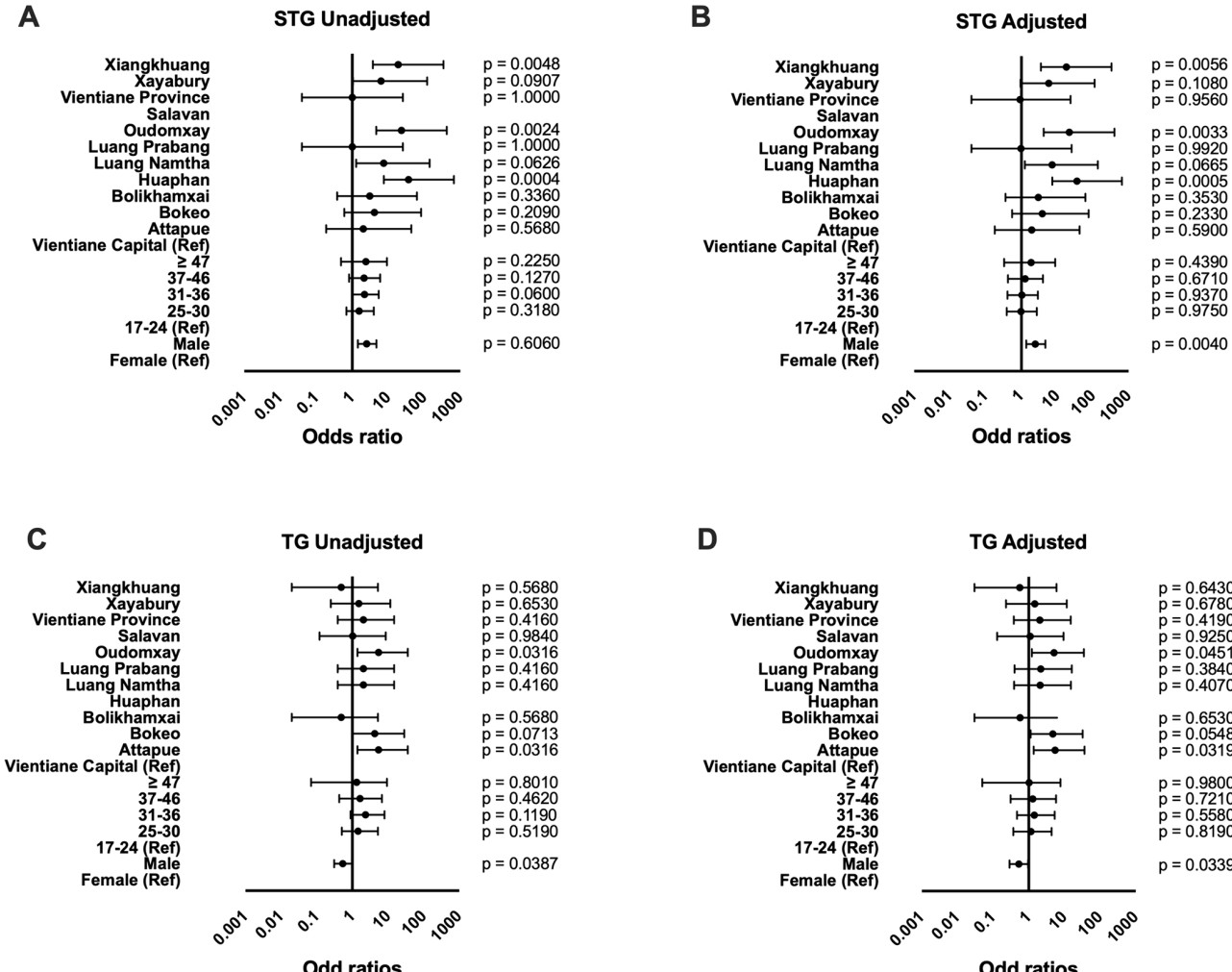

**Fig 5. Unadjusted (A, B) and adjusted (C, D) odds ratios for seropositivity using logistic regression analysis.** The variables that were entered were age group (17-24, 25-30, 31-36, 37-46 and ≥47 y) using the lowest age group as the reference, provinces (Vientiane Capital as reference). Odd ratios with 95% confidence intervals are displayed for scrub typhus group (STG) and typhus group (TG). No participants were seropositive for STG in Salavan, and no participants were seropositive for TG in Huaphan. Seropositivity was defined as immunofluorescence assay (IFA) titre ≥ 1:100 following enzyme-linked immunosorbent assay (ELISA) optical density (OD) ≥ 0.5. All participants with ELISA OD < 0.5 were not tested with IFA and were considered to be seronegative.

on behavioural or environmental risk factors, and this interpretation remains speculative. Further studies incorporating detailed exposure assessments are needed to clarify the underlying drivers of this gender difference. It is important to note that there were more male donors than female donors in the study. Seroprevalence studies conducted in healthy populations in neighboring countries, including North Vietnam [12] and Thailand [13], did not report gender-specific differences in seropositivity. A study conducted in Myanmar among patients who required blood sampling after medical consultation [14] demonstrated a pattern similar to our finding: scrub typhus was higher in men, whereas murine typhus group seropositivity group seropositivity was higher in women. We also found that participants aged 31–36 years had the highest percentage of STG IgG antibodies compared to the other age groups. In contrast, TG IgG antibodies were most prevalent among participants aged 37–46 years. However, there was no association with age group and seropositivity.

Huaphan, Oudomxay and Xiangkhuang were the three provinces with the highest STG seroprevalence. These northern provinces are predominantly mountainous and rural, with extensive forest cover and agricultural land use. Such environments are considered as suitable habitats for chigger vectors and rodent hosts, possibly contributing to higher transmission risk compared to more urban areas like Vientiane Capital and Luang Prabang, where STG seroprevalence was lower. However, there are currently no published data on the prevalence of *O. tsutsugamushi* in chigger mites from these provinces, and further entomological studies would be need to confirm this ecological hypothesis. We also found the background seroprevalence for STG in Luang Namtha for this study (7%) was similar to the prevalence of scrub typhus causing acute febrile illness in a previous study (6%) [1]. However, no STG seropositive cases were detected in Salavan in our selected population. This finding was surprising as a previous study on the causes of non-malarial febrile illness in Laos, conducted between 2008 and 2010, showed that 7% of patients were confirmed to have acute scrub typhus [1]. This might be explained by different exposure risks among blood donors compared to patients presenting to hospital with acute febrile illness. Relatively high TG seroprevalence was observed in both northern Laos (Oudomxay and Bokeo) and Southern Laos (Attapeu), although overall TG prevalence remained lower than STG. Notably, although Huaphan had the highest number of STG-seropositive participants, no TG-seropositive individuals were identified in this province.

A striking finding was that Vientiane Capital showed lower seroprevalence for both STG and TG compared to our previous report in 2010 [15], which reported STG IgG at 20.3% and TG IgG at 20.6%. In the current study, STG and TG were detected in only 1% and 2% of participants, respectively. Several factors may explain the conflicting findings between this study and previous reports, including that blood donors may not represent the general population, as they are less exposed to reservoirs or vectors, have limited age diversity, and differ in exposure-related behaviors. We did not collect detailed data on socioeconomic status, education level, occupation, or residential classification, these factors may influence differences in seropositivity between febrile patients and blood donors. While donor addresses were record, these may not reliably reflect long-term residence or exposure settings, as individuals may work or temporarily in different areas. In addition, residential classification (urban versus rural) was not systematically recorded for donors in the present study, limiting direct comparison of exposure setting with the previous report. Therefore, potential demographic and occupational selection bias among blood donors cannot be excluded. One possible explanation is that sample collection occurred during the COVID-19 pandemic, with blood collection between January and August 2021. During this period, tighter movement restrictions were implemented in April 2021 which may have temporarily influenced population mobility and short-term exposure. However, IgG antibodies can persist for at least 18 months in scrub typhus patients [16] and up to 12 months in murine typhus patients [9], showing higher antibody levels due to recent immune activation. In contrast, blood donors represent individuals with past exposure resulting in lower IgG titers. This difference in antibody persistence may explain the lower number of seropositive cases among healthy donors compared with patients. In addition, many areas in Vientiane Capital have become more urbanized over the past decade, leading to a shift in vector habitats toward rural areas and we also observed fewer scrub typhus cases in Mahosot Hospital.

This study has several limitations. First, we only surveyed twelve of seventeen provinces due to availability of samples. Second, we selected 100 participants per province, which may not accurately reflect the true exposure risk in each region. In addition, samples were obtained from blood donors, resulting in underrepresentation of older individuals and limiting the upper age range of participants. As selection within age strata was not randomized, the final age distribution reflects the characteristics of the donor population rather than the general population. These factors may introduce selection bias, and the relatively small sample size resulted in wide confidence intervals for several odds ratios, indicating substantial uncertainty in these estimates. Third the study was conducted during the COVID-19 pandemic, which may have influenced the resident population behaviours and local exposure risk. Finally, we performed IFA testing only on ELISA-positive samples (OD ≥ 0.5), assuming these participants had IgG titers < 1:100. Although initial screening by ELISA followed by IFA has been validated with high sensitivity (>95%) for both STG and TG [14] in this population, this approach may still have missed some seropositive individuals.

## Conclusion

This study demonstrates substantial prior exposure to both STG and TG in multiple provinces of Laos, including during the COVID-19 period. However, these findings reflect past exposure in healthy individuals and do not directly quantify the current burden of febrile illness attributable to rickettsia infections.

## Supporting information

**S1 Table. Median ELISA Optical Density (percentile; p25, p75) for scrub typhus group (STG), typhus group (TG) by participant demographic.**
(XLSX)

**S2 Table. Percentage of samples positive using ELISA OD ≥ 0.5 for rickettsia IgG as cut-off point with 95% confidence intervals for each gender, age group and region for STG and TG.**
(XLSX)

**S3 Table. Number of samples tested with IFA after screening with ELISA and considered seropositive out of total number of participants for gender, age group and region.** IFA samples were considered seropositive with a titre ≥ 1:100 for STG and TG.
(XLSX)

**S4 Table. Unadjusted and adjusted odds ratio of seropositivity for scrub typhus group (STG) using logistic regression analysis.**
(XLSX)

**S5 Table. Unadjusted and adjusted odds ratio of seropositivity for murine typhus group (TG) using logistic regression analysis.**
(XLSX)

**S1 Dataset. Raw data supporting the findings of this study.** The dataset includes raw ELISA (optical density, OD) and IFA (antibody titer) results. Seropositivity was defined as an IFA IgG titer ≥1:100 after initial ELISA screening with OD ≥ 0.5, indicating prior rickettsial exposure. Data are categorized by age, age group, gender, and province.
(XLSX)

## Acknowledgments

We are grateful to the director and staff of the Microbiology Laboratory, Mahosot Hospital, and the blood donors and staff who participated to earlier studies.

## Author contributions

**Conceptualization:** Audrey Dubot-Pérès, Matthew T. Robinson, Elizabeth A. Ashley.

**Data curation:** Weerawat Phuklia.

**Formal analysis:** Weerawat Phuklia, Mavuto Mukaka.

**Funding acquisition:** Elizabeth A. Ashley.

**Investigation:** Jantana Wongsantichon, Ampai Tanganuchitcharnchai, Stuart D. Blacksell.

**Methodology:** Jantana Wongsantichon, Ampai Tanganuchitcharnchai, Kaisone Padith, Stuart D. Blacksell.

**Resources:** Chantala Souksakhon, Mayfong Mayxay, Elizabeth A. Ashley.

**Supervision:** Elizabeth A. Ashley.

**Validation:** Jantana Wongsantichon.

**Writing – original draft:** Weerawat Phuklia, Elizabeth A. Ashley.

**Writing – review & editing:** Weerawat Phuklia, Jantana Wongsantichon, Mavuto Mukaka, Koukeo Phommasone, Mayfong Mayxay, Stuart D. Blacksell, Audrey Dubot-Pérès, Matthew T. Robinson, Elizabeth A. Ashley.

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
