## [Decision Letter · Decision Letter 0]

21 Jan 2026

Response to Reviewers
Revised Manuscript with Track Changes
Manuscript

Shaden Kamhawi

co-Editor-in-Chief

Paul Brindley

co-Editor-in-Chief

**Journal Requirements:**

1) Some material included in your submission may be copyrighted. According to PLOSu2019s copyright policy, authors who use figures or other material (e.g., graphics, clipart, maps) from another author or copyright holder must demonstrate or obtain permission to publish this material under the Creative Commons Attribution 4.0 International (CC BY 4.0) License used by PLOS journals. Please closely review the details of PLOSu2019s copyright requirements here: PLOS Licenses and Copyright. If you need to request permissions from a copyright holder, you may use PLOS's Copyright Content Permission form.

Potential Copyright Issues:

- Figures 1 and 4. Please (a) provide a direct link to the base layer of the map (i.e., the country or region border shape) and ensure this is also included in the figure legend; and (b) provide a link to the terms of use / license information for the base layer image or shapefile. We cannot publish proprietary or copyrighted maps (e.g. Google Maps, Mapquest) and the terms of use for your map base layer must be compatible with our CC BY 4.0 license.

2) Please amend your detailed Financial Disclosure statement. This is published with the article. It must therefore be completed in full sentences and contain the exact wording you wish to be published.

**Reviewers' comments:**

**Key Review Criteria Required for Acceptance?**

**Methods**

-Are the objectives of the study clearly articulated with a clear testable hypothesis stated?

-Is the study design appropriate to address the stated objectives?

-Is the population clearly described and appropriate for the hypothesis being tested?

-Is the sample size sufficient to ensure adequate power to address the hypothesis being tested?

-Were correct statistical analysis used to support conclusions?

-Are there concerns about ethical or regulatory requirements being met?

Reviewer #1: The methods are generally clear and appropriate, but please note the following:

1. It is stated that "concentrations of 1:8000, 1:4000 and 1:2000 dilutions of antigen" were used to coat plates, although normally this would be expressed as a protein concentration. If the investigators are not measuring protein content, how do they standardise the coating process between different batches of antigen?

2. The selection of samples per province was stratified by age and gender, but within these strata, was sample selection randomised and if so, how?

3. Details of the fluorescent conjugate used for the IFA are not provided. Please include.

4. In the Introduction, it is stated that GIS methods would be used to identify disease hotspots. This doesn’t seem to fit with the actual statistical approach employed?

Ethics and regulatory requirements seem to be met.

Reviewer #2: Are the objectives of the study clearly articulated with a clear testable hypothesis stated? Yes

-Is the study design appropriate to address the stated objectives? Yes

-Is the population clearly described and appropriate for the hypothesis being tested? Yes

-Is the sample size sufficient to ensure adequate power to address the hypothesis being tested? The issues in sample size have been discussed by the authors.

-Were correct statistical analysis used to support conclusions? Yes

-Are there concerns about ethical or regulatory requirements being met? No

**Results**

-Does the analysis presented match the analysis plan?

-Are the results clearly and completely presented?

-Are the figures (Tables, Images) of sufficient quality for clarity?

Reviewer #1: The presentation of the results is generally of sufficient quality with the following small areas for improvement.

1. The narrative doesn't always refer to the correct panels for Fig. 2 (only A and B are mentioned).

2. The ODs are provided as medians and percentiles it seems, but in inconsistent format (using a comma separator or dash). For instance, "The overall median OD percentage for STG was 0.034 (0.017, 0.170), and for TG it was 0.046 (0.020-0.162)." It is only when reading the supplement that it is clear the numbers in parentheses are percentiles rather than ranges. This needs to be made explicit in the main text. Also, these appear to be OD values and not "percentages".

3. The narrative highlights certain age groups as having a higher seroprevalence, but this isn't significant for either disease (or even close). So, it should simply be stated that seroprevalence was not associated with age bracket for either disease.

4. The labels on the plots in Fig. 4 are faint, low resolution and poorly legible. Please improve.

5. The panel label for Fig. 5D should read "adjusted".

Reviewer #2: Does the analysis presented match the analysis plan?- Yes

-Are the results clearly and completely presented? Yes

-Are the figures (Tables, Images) of sufficient quality for clarity? Yes

**Conclusions**

-Are the conclusions supported by the data presented?

-Are the limitations of analysis clearly described?

-Do the authors discuss how these data can be helpful to advance our understanding of the topic under study?

-Is public health relevance addressed?

Reviewer #1: The discussion and conclusions do a reasonable job, but could dig a bit deeper:

1. Are information available on any potential demographic biases among blood donors in Laos, in terms of things like socioeconomic status, education level, residence in more urbanised areas etc.? The authors hint at likely differences between ST/MT febrile patients and donors, but this could be explored a bit more. It seems unlikely that your average subsistence farmer is well represented among blood donors, for example.

2. The origin of the samples during the peak of the Covid pandemic is mentioned as a possible limitation. Could the Covid-19 pandemic policies of Laos be briefly summarised here? If they were as restrictive as Vietnam's zero-Covid approach for example, the impact on exposure to ST/MT could have been very substantial for many months at least.

3. The gender differences in seroprevalence are barely touched upon. Are there comparable data for surrounding countries?

4. It is stated that "However, no STG seropositive cases were detected in Salavan in our selected population. This finding was surprising as a previous study on the causes of nonmalarial febrile illness in Laos showed that 7% of patients were confirmed to have acute scrub typhus [1]." The age of the cited study should be highlighted here, as no doubt the economic/demographic changes in Laos in the past 10 - 20 years have been seismic.

5. The conclusions state “...confirming that rickettsial infections are widespread and remain an important cause of febrile illness throughout the country”. I don't think this study can reach this conclusion, as it is examining exposure in healthy individuals some years ago (and during the pandemic). It can only conclude that exposure to both diseases seems substantial in healthy individuals, even during a period when exposure would have been expected to be impeded by movement restrictions.

Reviewer #2: Are the conclusions supported by the data presented? Yes

-Are the limitations of analysis clearly described? Yes

-Do the authors discuss how these data can be helpful to advance our understanding of the topic under study? Yes , I have added some comments under "Summary and General Comments"

-Is public health relevance addressed? Yes

**Editorial and Data Presentation Modifications?**

Reviewer #1: 1. In the method section "Blood donors", the sentence "Serum samples were collected at the National Blood Centre and stored at -30°C before being transferred to the Lao-Oxford-Mahosot Hospital-Wellcome Trust Research Unit (LOMWRU), where they were stored at -80°C," should end in a full stop.

2. In the legend for Fig. 3, replace "represent" with "represented".

3. The subtitle "Logistic regression seropositivity" in the Results should be "Logistic regression and odds of seropositivity".

4. "By province (reference: Vientiane Capital)” is repeated at the beginning of a sentence in the paragraph under the "Logistic regression seropositivity" subheading. (Lack of line numbers in the manuscript is making these corrections difficult to describe...!)

5. The abbreviation "aOR" for adjusted odds ratio is not used consistently.

6. At the beginning of the Discussion, "Rickettsia infection" should be "rickettsial infections".

7. The manuscript refers to either 17 or 18 provinces of Laos in total. Please check the correct number.

Reviewer #2: I have added some comments under "Summary and General Comments".

**Summary and General Comments**

Reviewer #1: This is an important study on the seroprevalence of scrub typhus and murine typhus among blood donors in Laos, providing a new perspective following some groundbreaking work on these infections in Laos that is now more than 10 years old. Generally, the manuscript is straightforward and interpreted appropriately, but there are some stylistic problems, minor methodological issues, and the discussion requires a bit more context. See specific points to be addressed in the boxes above.

The results part of the abstract is rather packed with statistical reporting, which alongside the use of abbreviations at the beginning of sentences, makes it a very tough read. It would be more useful to summarise the key findings, simply stating what was significantly different or not, without all the odds ratios and CIs being quoted. Ironically, one of the more interesting findings (the significant and opposite gender difference for each disease) is not even mentioned here.

Reviewer #2: Comments on “Scrub and murine typhus seroprevalence among blood donors in Laos”

1. Abstract: “scrub typhus” instead of “scrub”.

2. Why were only scrub typhus and murine typhus selected? Why was seroprevalence to spotted fever rickettsia also not tested for? In many studies, spotted fever rickettsial infections are more prevalent than murine typhus.

3. Geographically and ecologically, how would the authors describe Huaphan, Oudomxay and Xiangkhuang? These places have higher seropositivity than that is the capital city.

4. No correlation with age. 17 to >47 years was tested.

5. The difference between the current and previous reported seroprevalence is very striking. The hypothesis that COVID restrictions might have had role to play may be the most like explanation. The actual seroprevalnece may be much more. This may be discussed in the paper.

Additionally, the previous study (12) had enrolled participants from urban and peri-urban locations around the capital area. Were the blood donors selected in the current study from urban or rural areas, is this information available?

6. What could explain the higher seropositivity for MT in females? A few lines on this may be added to the Discussion.

7. These infections re sympatric in many areas. How many subjects had seropositivity for both infections?

PLOS authors have the option to publish the peer review history of their article (what does this mean? ). If published, this will include your full peer review and any attached files.

**Do you want your identity to be public for this peer review?** For information about this choice, including consent withdrawal, please see our Privacy Policy .

Reviewer #1: No

Reviewer #2: **Yes:** Manisha Biswal

**Figure resubmission:**

**Reproducibility:** To enhance the reproducibility of your results, we recommend that authors of applicable studies deposit laboratory protocols in protocols.io, where a protocol can be assigned its own identifier (DOI) such that it can be cited independently in the future. Additionally, PLOS ONE offers an option to publish peer-reviewed clinical study protocols. Read more information on sharing protocols at https://plos.org/protocols?utm_medium=editorial-email&utm_source=authorletters&utm_campaign=protocols

---

## [Editor Report · Decision Letter 1]

18 Feb 2026

Dear Dr Phuklia,

We are pleased to inform you that your manuscript 'Scrub and murine typhus seroprevalence among blood donors in Laos' has been provisionally accepted for publication in PLOS Neglected Tropical Diseases.

Best regards,

Joseph M. Vinetz

Section Editor

Joseph Vinetz

Section Editor

Shaden Kamhawi

co-Editor-in-Chief

Paul Brindley

co-Editor-in-Chief

---

## [Editor Report · Acceptance letter]

Dear Dr Phuklia,

We are delighted to inform you that your manuscript, "Scrub and murine typhus seroprevalence among blood donors in Laos," has been formally accepted for publication in PLOS Neglected Tropical Diseases.

Best regards,

Shaden Kamhawi

co-Editor-in-Chief

Paul Brindley

co-Editor-in-Chief
